# Development and delivery of an allied health team intervention for older adults in the emergency department: A process evaluation

**Marica Cassarino**[1,2]*, **Úna Cronin**[3,4], **Katie Robinson**[2], **Rosie Quinn**[5], **Fiona Boland**[6], **Marie E. Ward**[7], **Rosa McNamara**[8], **Margaret O'Connor**[9,10], **Gerard McCarthy**[11], **Damien Ryan**[4,10], **Rose Galvin**[2]

**1** School of Applied Psychology, University College Cork, Cork, Ireland, **2** School of Allied Health, Faculty of Education and Health Sciences, Ageing Research Centre, Health Research Institute, University of Limerick, Ireland, **3** Rapid Innovation Unit, Graduate Entry Medical School, Health Research Institute, University of Limerick, Limerick, Ireland, **4** Retrieval, Emergency and Disaster Medicine Research and Development Unit (REDSPoT), Emergency Department, University Hospital Limerick, Dooradoyle, Limerick, Ireland, **5** Emergency Department, Our Lady of Lourdes Hospital Drogheda, Drogheda, Ireland, **6** HRB Centre for Primary Care Research, Royal College of Surgeons in Ireland, Dublin, Ireland, **7** School of Psychology, Trinity College, The University of Dublin, Dublin, Ireland, **8** Emergency Department, St. Vincent University Hospital, Dublin, Ireland, **9** Department of Ageing and Therapeutics, University Hospital Limerick, Dooradoyle, Limerick, Ireland, **10** Graduate Entry Medical School, Faculty of Education and Health Sciences, University of Limerick, Limerick, Ireland, **11** Emergency Department, Cork University Hospital, Cork, Ireland

* mcassarino@ucc.ie

## Abstract

### Background

There is encouraging evidence that interdisciplinary teams of Health and Social Care Professionals (HSCPs) can enhance patient care in the Emergency Department (ED), especially for older adults with complex needs. However, no formal process evaluations of implementations of ED-based HSCP interventions are available. The study aimed to evaluate the development and delivery of a HSCP team intervention for older adults in the ED of a large Irish teaching hospital.

### Methods

Using the Medical Research Council (MRC) Framework for process evaluations, we investigated implementation and delivery, mechanisms of impact, and contextual influences on implementation by analysing the HSCP team's activity notes and participant recruitment logs, and by carrying out six interviews and four focus groups with 26 participants (HSCP team members, ED doctors and nurses, hospital staff). Qualitative insights were analysed thematically.

### Results

The implementation process had three phases (pre-implementation, piloting, and delivery), with the first two described as pivotal to optimise care procedures and build positive stakeholders' involvement. The team's motivation and proactive communication were key to

**Data Availability Statement:** In line with the restrictions imposed by the Health Service Executive (HSE) Mid-Western Regional Hospital

Research Ethics Committee, which approved the ethics of this study, data collected in this study cannot be shared publicly because participants can be identifiable given their role in the setting of the study, and because the participants did not give consent to have their responses made publicly available. Data are available from the Health Service Executive (HSE) Mid-Western Regional Hospital Research Ethics Committee (ULHGResearchEthicsandClinicalTrials@hse.ie) for researchers who meet the criteria for access to confidential data.

**Funding:** This research was supported by the Health Research Board of Ireland through the Research Collaborative for Quality and Patient Safety (RCQPS 2017-2) awarded to Prof. Rose Galvin. The sponsor is not involved in the design of the study and collection, analysis, interpretation of data, or in writing the manuscript.

**Competing interests:** The authors declare that they have no competing interests

promote acceptability and integration in the ED (Theme 1); also, their specialised skills and interdisciplinary approach enhanced patient and staff's ED experience (Theme 2). The investment and collaboration of multiple stakeholders were described as essential contextual enablers of implementation (Theme 4). Delivering the intervention within a randomised controlled trial fostered credibility but caused frustration among patients and staff (Theme 3).

## Discussion

This process evaluation is the first to provide in-depth and practical insights on the complexities of developing and delivering an ED-based HSCP team intervention for older adults. Our findings highlight the importance of establishing a team of HSCPs with a strong interdisciplinary ethos to ensure buy-in and integration in the ED processes. Also, actively involving relevant stakeholders is key to facilitate implementation.

## Trial registration

ClinicalTrials.gov, NCT03739515; registered on 12th November 2018.

## Introduction

Identifying effective quality improvement strategies in Emergency Departments (EDs) has become increasingly urgent to address growing numbers of ED attendances and their negative impact on patient and process outcomes [1]. The promotion of interdisciplinary team care and the introduction of Health and Social Care Professionals (HSCPs) to the ED have been highlighted as promising strategies to enhance the care of patients with complex health and social care needs including older adults and those with multimorbidity [2–6]. Older adults are a particularly vulnerable cohort as they are frequent ED users and at high risk of adverse outcomes following ED visit [7, 8].

HSCP teams operating in the ED are routine practice in some countries, such as Australia [9], but represent a new and underexplored model of care in other contexts, as noted in a recent systematic review [10]. Thus, robust evidence on the effectiveness of ED-based HSCP interventions is scarce and evaluations of the implementation of HSCP team interventions for older adults in the ED are currently limited to analyses of patient and staff satisfaction [11–13]. Exploration of stakeholders' views on ED models of care have enabled the identification of operational and relational factors that can influence the successful implementation of new interventions [14–17].

However, implementation frameworks for health service interventions suggest consistently that healthcare change is a complex process influenced by factors at multiple levels, ranging from individuals, to the characteristics of the intervention itself, to the operational and relational features of the inner and outer context of the intervention [18–20]. Therefore, comprehensive and structured investigations are needed to gain a better understanding of the complex mechanisms and determinants of change for new quality improvement strategies [19, 21–23]. Process evaluations have increasingly become instrumental to capture such complexity [24, 25], offering practical insights on the development and delivery of new models of care [18, 26, 27].

Building on the Medical Research Council (MRC) Framework for process evaluations of complex interventions [27], the present study aimed to provide an in-depth evaluation of the process of developing and delivering the OPTI-MEND (Optimising early assessment and

intervention by health and social care professionals in the emergency department) randomised controlled trial (ClinicalTrials.gov, NCT03739515)—a HSCP team intervention where an ED-based team composed of a senior medical social worker, a senior occupational therapist, and a senior physiotherapist provided early assessment, intervention and ED discharge plans to adults aged ≥65 years. The characteristics and theoretical underpinnings of OPTI-MEND trial are described in detail in the study protocol [28] as well as the report of clinical effectiveness [29]. A cost effectiveness evaluation is in preparation at the time of writing.

The overall aim of this process evaluation is to provide a comprehensive account of the development and delivery of this ED-based HSCP intervention, and it has three main objectives:

1. To describe and analyse the implementation of the HSCP intervention, in terms of the processes occurred to develop and deliver the intervention, the level of fidelity/adaptations, and the dose and reach;

2. To explore the mechanisms of impact within the intervention, including individual, operational and relational mediators, and unexpected pathways and effects;

3. To identify key contextual factors of delivery and impact at the levels of individuals, the ED (physical environment, operations and relations), the hospital, the community and the wider healthcare system.

For the purpose of this study, we considered "impact" as the successful implementation of the team within the ED rather than the clinical or cost effectiveness of the intervention, which have been investigated separately [29].

## Materials and methods

### Design

As recommended within the MRC framework [27], this process evaluation integrated qualitative and quantitative methods in the attempt to capture both the quality and quantity of the intervention. The study protocol is available elsewhere [28]. While no specific guidelines are available for the reporting of process evaluations [27], this study was reported in accordance with the Standards for Reporting Implementation Studies (STARI) statement [30] and the Criteria for Reporting the Development and Evaluation of Complex Interventions in Healthcare (revised guideline CReDECI 2) [31], where appropriate. A full STARI statement can be found in the S1 Table.

### Context

The process evaluation was conducted in the ED of a hospital with a large catchment area in the Mid-West Region of Ireland, where the intervention was carried out; the ED provides acute care to over 60,000 adults and children a year. Older adults represent approximately 30% of presentations in this setting [32] and are more likely to present with a complexity of functional, medical and psychosocial issues. This cohort were identified as a target group who were most likely to benefit the most from this model of care in our extensive stakeholder engagement meetings prior to undertaking the RCT [17]. The intervention consisted of a dedicated team of HSCPs providing early assessment and intervention to older people aged ≥65 years presenting to the ED with complex care needs. The impact of the intervention when compared to usual care was tested on our two primary outcomes of ED length of stay and hospital admission rates. Secondary outcomes focused on a range of other patient, process, and clinical outcomes.

## Participants

The study involved 26 participants (57% female) who took part in four focus groups (n = 20) and 1:1 interviews (n = 6). The sample included:

- the three HSCPs composing the team who carried out the intervention (one senior medical social worker, one senior occupational therapist, one senior physiotherapist);

- 10 ED doctors;

- 10 ED nursing staff members, including and the research nurse employed in the OPTI-- MEND trial;

- three hospital staff members, including two Heads of Department and one Data Management officer, who contributed to the development and implementation of the intervention.

The four focus groups comprised the HSCP team, eight ED doctors, and nine nurses. The research nurse involved in the trial, two ED doctors, and the three hospital staff members completed 1:1 interviews.

The participants were recruited through purposive, convenience and snowball sampling, with the clinical and research team acting as gatekeepers; study leaflets were also distributed in the ED. All participants were provided with an information sheet outlining the aims and methods of the study and all signed a written informed consent. Ethics approval for the study was received from the HSE Mid-Western Regional Hospital Research Ethics Committee (REC 103/18) in September 2018.

## Explored domains

In line with the study objectives and protocol [28], the domains of interest for this process evaluation included the implementation of the project, the mechanisms of delivery and impact, and the contextual influences on the intervention, as shown in Table 1. A detailed account of data sources and analysis types for each domain are available in the study protocol [28].

**Table 1. Process evaluation domains.**

| Objective | Domain | Gathered data |
|---|---|---|
| 1. Describe and analyse implementation | Process | Activities, inputs, structures, and resources needed to develop and deliver the intervention |
| | Fidelity and adaptations | • Adherence to the intervention protocol and to evidence on national and international practice [18],<br>• Intervention adaptations based on Stirman's framework [33] |
| | Dose | Duration, intensity, and frequency of the intervention |
| | Reach | Proportion of eligible ED patients who took part in the intervention |
| 2. Explore the mechanisms of impact within the intervention | Participants | Interactions with, and reactions to, the intervention by ED patients and staff |
| | Mediators | Individual, relational and operational facilitators and barriers |
| | Unexpected pathways and consequences | Changes in practices and procedures |
| 3. Identify key contextual factors of delivery | Intervening contextual influences | Potential influences at the level of:<br>• ED of intervention;<br>• hospital;<br>• community;<br>• healthcare system |

Notes. ED = Emergency Department

To avoid confusion, the following definitions were developed and agreed by the research team and are used hereafter:

- *Stakeholder*: Any individual who interacted with and/or facilitated the implementation of the HSCP team intervention in the ED, hospital staff; clinicians outside the hospital; researchers involved in the project.

- *Participant*: Any individual who took part in the process evaluation. This group may include stakeholders.

## Data collection and analysis

Data collection for this study took place between June and July 2019. The main form of data collection for this study was qualitative, through focus groups and semi-structured interviews, to explore the direct experiences of individuals who had carried out the intervention or interacted with it in the ED. Using both focus groups and interviews enabled better accommodation of participants' needs with regards to their availability and their preference for sharing their experience with others or individually. Qualitative data regarding the development and delivery of the intervention were integrated with quantitative data and notes from the recruitment and activity logs to develop a description of fidelity, dose, and reach.

A full interview schedule is included in the study protocol [28]; this was used as a guide to ensure that the three areas of interest of our investigation (implementation, mechanisms and context) were covered during the interview. The interviews and focus groups were audio recorded for verbatim transcription. One member of the research team (MC), who has experience in qualitative methods in the area of health, transcribed and analysed the qualitative data using the software NVivo version 11 Plus (QSR International Pty Ltd). Another researcher (RG) contributed to the data analysis and interpretation. The transcribed data were analysed iteratively and inductively, guided by the MRC framework, in accordance with the six steps of thematic analysis [34, 35]. In line with Saunders et al. [36], data saturation was agreed by the two researchers (MC and RG) during data collection when it was observed that new data repeated what had been already expressed in previous responses.

In conducting the analysis, reflexivity was achieved by considering the researchers' background and past experiences within the setting of the study (i.e., previous ED service user) in the design and conduct of the study. Further strategies to reduce response or analysis biases included the use of the MRC framework, as well as sharing the design and analysis of the study with the full research team.

In accordance with Data Protection policies, the study participants provided written informed consent to sharing anonymised extracts of interview transcripts; access to the full dataset is limited to the research team members who collected and analysed the data (MC and RG).

## Findings

An overview of the implementation of the intervention together with the mechanisms and context of impact is presented in Fig 1. In this section, we firstly describe and quantify the implementation of the HSCP intervention, and then discuss key themes related to the mechanisms and contextual factors of impact.

## Process–developing the HSCP team intervention in the ED

A detailed account of the implementation process is included in the S1 File. As shown in Table 2, the process of implementing the HSCP intervention was structured in three main

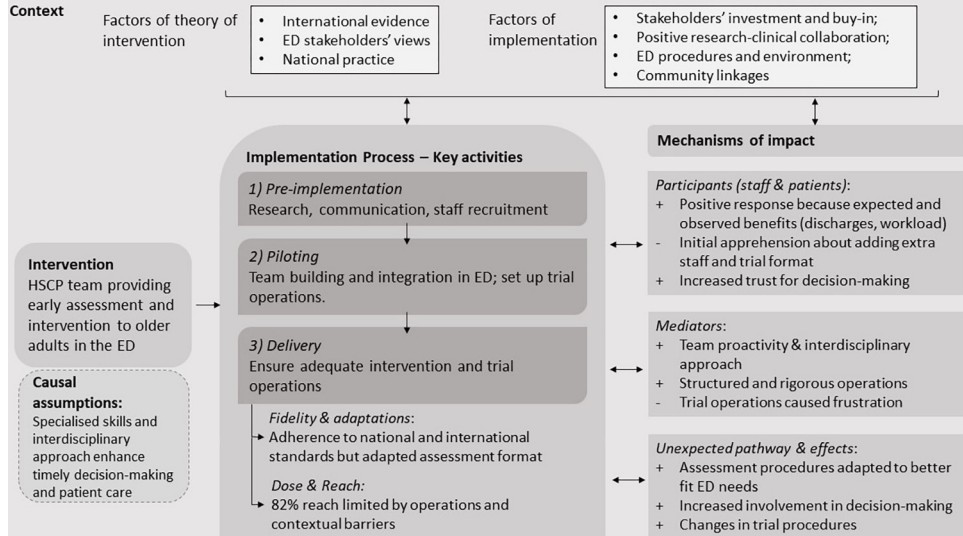

**Fig 1. Implementation framework of a HSCP team intervention for older adults in the ED [27].** (+) indicates an enabler, (-) refers to a barrier.

phases. In the pre-implementation phase, research and communication activities were crucial to establish a clear evidence base to guide the intervention, to determine the structures and the supports needed for a successful implementation, to raise awareness and acceptability for the intervention among ED and hospital staff, and to identify the criteria for recruiting the most appropriate staff in the HSCP team.

The piloting phase, after recruitment of the three HSCPs (one Senior Medical Social Worker, one Senior Occupational Therapist and one Senior Physiotherapist) and space allocation in the ED, was crucial for the team to develop an interdisciplinary approach informed by existing models of care, get integrated into the ED environment and identify the most optimal care

**Table 2. Implementation process.**

| Phase | Duration | Key activities and aim | Inputs, resources, and structures |
|---|---|---|---|
| Pre-implementation | 12 months (before HSCP team recruitment) | Research: Establish evidence to guide intervention | • Systematic review<br>• Stakeholders' input<br>• ED data flow analysis |
| | | Communication:<br>• Set up structures and supports<br>• Enhance acceptability<br>• Recruit appropriate staff | • Input from key ED, hospital, and healthcare service staff |
| Piloting | Six weeks (after HSCP team recruitment) | Team building and integration into ED environment:<br>• Define team procedures<br>• Allocate adequate space and equipment | • Team internal communication<br>• Meetings with the ED medical and nursing staff<br>Establishment of patient care pathways and criteria |
| | | Set up trial operations | • Liaison between HSCP team, research team, and hospital IT support |
| Delivery | Six months (Dec 2018-May 2019) | Ensure adequate intervention operations | • Interprofessional care models<br>• Positive relationships within the ED |
| | | Ensure adequate trial operations | • Liaison between HSCP team and research team<br>• Ongoing engagement with key ED, hospital, and healthcare service stakeholders |

Notes. ED = Emergency Department; HSCP = Health and Social Care Professional; IT = Information Technology

pathways and criteria. At the same time, liaison between the HSCP team, research team and support departments in the hospital enabled the intervention to work effectively within a trial.

During the intervention delivery (phase 3), the ongoing engagement between the HSCP team and other relevant stakeholders and parties was fundamental to ensure the appropriate operations both for the intervention and the trial.

### Intervention delivery: Fidelity, dose, and reach

A detailed description of the assessment and interventions delivered by the HSCP team can be found in the trial report [29]. The target population for the intervention were older adults aged ≥65 years with a variety of index complaints; further details on patients' inclusion criteria are described in detail in the trial report [29]. As part of the intervention, patients received holistic assessment by one or more members of the HSCP team with regards to their mobility, functional, cognitive, and psychosocial abilities. The team operated following an interdisciplinary approach whereby procedures were agreed as a team and decisions were made by a key case worker. Based on the results of the assessment, interventions and individualised discharge care plans were prescribed by the HSCP team in collaboration with the ED medical and nursing team.

The main findings related to fidelity, dose and reach are described in Table 3, and a detailed account can be found in S1 File.

In terms of fidelity, the HSCP team felt that that engagement with academic evidence and existing models of interdisciplinary care were critical to ensure that the content of the assessment and the target population were in line with international and national healthcare standards. However, adaptations to the assessment procedures and screening documentations were identified during the piloting phase, and communication with ED and hospital staff was pivotal to optimise care procedures (see S1 File for further details). These adaptations helped the team to share with the medical team a more comprehensive analysis and clearer recommendations for the patients, and thus enabled more timely care decisions.

Considering dose, the intervention was delivered during normal business hours, with 176 out of 214 eligible patients (82%) being included in the intervention. The team's operating hours were described by many participants, including the HSCP team, as an important factor of dose and reach. For instance, anecdotally the HSCP team members reported that many eligible patients were GP referrals presenting in the afternoon, but these could not be picked up by the team due to the 5pm finish time. Participants identified important contextual barriers to this, including the limits of the scope of practice of allied health professionals, as well as the absence of community services outside normal business hours, which would hinder the ability of a HSCP team to discharge the patient safely.

**Table 3. Intervention delivery.**

| Delivery dimension | Key findings | Enablers/barriers |
|---|---|---|
| Fidelity–adherence | Intervention delivered in line with international ED-based HSCP practice [10] and national standards [39] | • Engagement with academic evidence and models of interdisciplinary care |
| Fidelity—Adaptations | Assessment procedures and screening documentation adapted to highlight the analysis and recommendation made by the team | • Piloting period crucial to tailor procedures and inclusion criteria.<br>• Communication and engagement with ED/hospital staff and research stakeholders useful to optimise adaptations |
| Dose | Team operating Monday to Friday, 8am-5pm<br>Mean number of patients seen daily = 1.82, SD = 0.93, range = 1–5<br>Approximately four-five hours interaction with patient | • Team's operating hours limited dose and reach<br>• Absence of community services outside business hours limited HSCP team's opportunities to discharge the patient safely |
| Reach | 176/214 (82%) eligible patients reached | • Age inclusion criterion (65+) useful to ensure good patient flow but limited reach to younger patients who may benefit from intervention<br>• Trial operations as both enabler and barrier (see Theme 3) |

Notes. ED = Emergency Department; HSCP = Health and Social Care Professional; SD = Standard Deviation.

Furthermore, while the patient age inclusion criterion (≥65 years) had been chosen to reach older adults who could benefit the most from the comprehensive assessment provided by a team of healthcare professionals [37, 38], it was felt that this may have limited reaching patients younger than 65 years old who would benefit from engaging with the HSCP team.

Lastly, an important factor of reach and dose was the delivery of the HSCP intervention within a randomised controlled trial, which is discussed in detail in section "Theme 3".

## Mechanisms and context of impact: Key themes

The participants' views on the HSCP intervention were overall positive, identifying benefits of the ED-based HSCP team for patients in terms of safer discharges and for staff members as to reduced workload and increased confidence in decision-making.

Four main themes were identified based on the accounts provided by the participants in this study:

1. The team's motivation and proactive communication promoted acceptability and integration in the ED;

2. The team's specialised skills and interdisciplinary approach enhanced patient and staff's ED experience;

3. The project stakeholders' investment was a key enabler of implementation and acceptability;

4. Using a trial format promoted credibility but caused frustration among patients and staff.

A table of quotes for each theme is included in S2 File.

**Theme 1: The team's motivation and proactive communication promoted acceptability and integration in the ED.**　　When discussing what aspects of the intervention worked well, most participants pointed at the HSCP team's motivation and investment in the intervention that enabled them to work effectively as an interdisciplinary team and to establish positive relationships with other ED staff members and the patients. ED staff members felt that the team's personal investment was highly reflected in their proactivity and flexibility while getting integrated in the ED operations, which were key to promote buy-in and acceptance.

> *"They were proactive, already having identified patients and identified issues. They'd be part of handover as well and they have identified people who may need their service. They have already initiated the actual work and then feed back to me, which is fantastic; you are not actually looking for them, they are looking for work and they are initiating it. So, I found them excellent"* (ED nurse 5)

Participants also valued the team's "open door" approach to communication with patients, ED staff members and community services, which fostered trust, and helped to optimise the intervention procedures as well as negotiate space within the complex ED environment:

> *"They were very capable of making their presence felt in a positive way and interacting always in a positive manner with the department staff"* (ED doctor 9)

From the HSCPs' perspective, the team felt that the structured processes of the ED and its relational culture of open feedback and mutual support facilitated setting up the intervention in an optimal manner within the short timeframe of the piloting phase:

> *"I think it's a good environment to. . .what you could do there in four weeks might take you eight to 12 weeks in a ward environment because you have a defined team and a defined*

*closed environment, so it's very conducive to feedback, communication and testing because there is a real process, a flow there, you can see end to end from triage to once they leave"* (HSCP team member 1)

Open and constructive communication was thus a key enabler of implementation, both within the ED and with relevant stakeholders, which helped to raise the profile of the team in the acute setting:

*"Keeping key people informed along the way. We had to demonstrate in abundance that we knew what we were doing and were invested in the project. We were open not only for the small things but also significant issues. If we didn't have that piece, I still think we would have done exactly as we did, it might have been slower to get off the ground."* (HSCP team member 3)

Despite these positive views, some ED staff members who started working in the ED during the delivery phase (i.e., after the HSCP had been introduced to the ED) were less aware of the intervention and felt that they would have benefited from being better informed. Also, some ED staff members mentioned that having ways to contact the team easily (e.g., bleep) could have increased accessibility in case the team was needed but they weren't reachable in their dedicated space.

**Theme 2: The team's specialised skills and interdisciplinary approach enhanced patient and staff's ED experience.**   The HSCP team's interdisciplinary approach was also an important strength of the intervention:

*"We are looking at the same person but with different perspectives, with different methods. And it's also respecting the other persons in the group and take on board that we all have different ways as long as the assessment is kind of standardised in a way that we are still looking at the same model, and it doesn't matter how you go about that."* (HSCP team member 1)

ED staff members further supported this view of interdisciplinary and holistic care from a team of professionals with specialised skills as crucial to increase patient satisfaction and their confidence about being discharged home, thus promoting admission avoidance. Importantly, having specialised HSCPs in the ED resulted in a reduced workload for ED staff members:

*"I think, for the patients themselves, whether it was getting a specific intervention, getting community services involved, getting an aid to help them, getting social work stuff happening in the community. I think it's all very beneficial stuff for patients. And it can be difficult for busy doctors and nurses to sort that out on any given day. Whereas you have somebody dedicated who knows the ins and outs of the community services, knows how to access stuff in a more straightforward manner. It's much more efficient"* (ED Doctor 2)

ED staff members also discussed direct benefits for them of having an interdisciplinary team of HSCPs, particularly with regards to increased awareness of care options for the patients and higher confidence in making decisions about the patient's care plan.

The HSCP team acknowledged that some level of negotiation with some of the ED medical staff was required initially to promote a less "risk avoidant culture" with regards to discharges, given the older frail population:

*"There was a lot of pressure to build that trust with the doctors, because it wasn't the norm and there wasn't access to it, patients would be sometimes admitted even though there was no need."* (HSCP team member 2)

While the team's skills and interdisciplinary approach were major strengths of the intervention, some hospital staff felt that this model of care could be further enhanced by building more integrated packages of care including home visits, review clinics or geriatric specialist services in the community, as well as considering the recruitment of HSCPs with advanced scope of practice who could make decisions without the need of a doctor's sign off:

*"Probably a clinical specialist role would have more clinical decision-making powers. As a profession we need to start looking at seeing patient in a primary level, because that's where's the real benefit, rather than waiting for the referral. That would free up the medical staff while utilising the hours for the benefit of the patients"* (Hospital staff 2)

**Theme 3: The project stakeholders' investment was a key enabler of implementation and acceptability.** As highlighted by many participants, having the investment and motivation of the hospital and academic stakeholders who valued the intervention and who collaborated positively with each other was crucial the positive development and implementation of the HSCP intervention in the ED; not only it provided valuable inputs to establish effective procedures based on existing practice and evidence, but also it facilitated the team's recruitment, sourcing space and equipment in the ED, providing technical infrastructure and support, and enabling the team to get integrated in the ED:

*"I was very excited by it, enthused by it. I just thought for me it was a case of 'we need to have these disciplines in there; if I can use research to prove how effective they are, it's a win-win'"* (Hospital staff 1)

Notably, having the back up of motivated stakeholders in the healthcare setting was instrumental to overcome the initial apprehension within the ED and the hospital site with regards to the practical implications of adding extra staff to the ED and employing a trial format.

Some participants reported initial concerns about whether having additional professionals in the ED might confuse older patients and be detrimental for patient/staff, as well as the risk of increasing demands on the ED from the community given the awareness that extra services are available, particularly for older frail patients:

*"I suppose, there is always a danger when services are created that they themselves create work. I think that that's a positive thing in that circumstance, but I suppose we always have a little fear that our community-based colleagues might become aware that the service is available within the ED for specific cohorts of patients and then decide to send patients to the ED because they know that this is available"* (ED doctor 10)

Initial apprehension was also recalled with regards to the impact of having a new interdisciplinary model of care on the scope of practice of the specific allied professions. While the team's skills and approach described in Theme 1 and 2 facilitated addressing these concerns during the delivery of the intervention, it was felt that building positive relations within the ED and the optimisation of the intervention procedures would have been more challenging without the support of multiple stakeholders within and outside the hospital. Overall, it was felt

that the development of the intervention provided a positive learning experience and proto-type for future collaboration between academic and healthcare bodies:

*"There never has been that collaboration between this kind of hospital side of it and the aca-demics side, let alone also bringing in kind of the IT element of it as well. So, I think there is a good bit of learning around that alright in terms of how we can all play together nicely"* (Hos-pital staff 3)

**Theme 4: Using a trial format promoted credibility but caused frustration among patients and staff.** From an operational viewpoint, delivering the HSCP team intervention within a randomised controlled trial bore both benefits and challenges for the team.

Firstly, having research staff and technical support allocated for the trial data collection and management facilitated the accuracy and integrity of the research side of the project while let-ting the HSCP team focus on the intervention. However, some adaptations to the trial proce-dures were required along the way to better fit the needs of the intervention. These included 1) allowing all members of the HSCP team and the research nurse to consent patients to better meet the practicalities of staff availability (e.g., the research nurse being on leave), and 2) mov-ing the research nurse from the HSCP room to a separate space to ensure privacy for the team and patients as well as reducing space demands in terms of extra equipment for safe data col-lection and storage. These adaptations received full ethical approval before being implemented.

Secondly, while the comprehensive baseline assessment carried out within the trial enabled the team to gather extra information about the patient, it was felt that the time needed for the assessment and the consenting and randomisation procedures delayed the intervention and thus limited dose and reach:

"*Providing information to the patient, going through all the background information with them, giving them time to read through it and understand and then going back and ask whether they consented or not. That was something that I suppose added a bit of time to the assessment. That kind of delayed things a bit*" (HSCP team member 1)

From a relational point of view, both the HSCP team and other participants felt that intro-ducing the intervention within a randomised controlled trial was beneficial because motivated them to be rigorous in their practice and increased acceptability:

*"I think that it can be very hard to convince health care management people to fund a new ser-vice in the current language in the absence of definitive evidence. So, I think having this worked particularly well, having a research-funded team coming in and showing benefit, hav-ing an impact immediately makes a very positive case for doing further interventions in this manner."* (ED doctor 8)

On the other hand, some participants felt that the trial format had at times negative impact in how patients and staff members reacted to the intervention, in particular randomisation to the control group:

*"It's very difficult to tell somebody that they were in the control group after they have spent some much time giving you so much information, and they really want support and informa-tion about other supports in terms of access and assessment. Some of them became quite dis-tressed at times, when they were in the control group"* (HSCP team member 2)

## Discussion

### Summary of findings

This mixed-methods process evaluation aimed to describe the implementation and factors of impact of an ED-based HSCP team intervention providing early assessment and interventions to older adults with frailty and complex needs.

Considering the implementation process, the two phases preceding the intervention delivery (i.e., pre-implementation and piloting) were described as essential to the successful implementation of the intervention because enabling the gathering of inputs from relevant stakeholders, effective preparation, team building and buy-in from existing staff. The content and target population of the intervention aligned with existing evidence and practice [11, 12, 40], but some operational adaptations were necessary (e.g., assessment tool) to tailor the team's procedures to the needs and dynamics of the ED. The team's operating hours and patient inclusion criteria were perceived as limiting dose and reach, but at the same time they enabled the team to identify the patients most in need and dedicate enough time for an in-depth and comprehensive assessment.

In terms of mechanisms of impact, our participants felt that the HSCP team's motivation and relational skills enabled them to get easily accepted and integrated in the ED team; also, their specialised skills and interdisciplinary approach contributed to enhance the ED experience of both patients and other staff members, promoting safer discharges, patient-centred care, reduced workload for the medical team and confidence in decision-making. Notably, our participants expressed positive views without knowing objective results on the intervention outcomes (i.e., trial results not available at the time of data collection), which highlights a positive impact of the intervention beyond clinical and cost effectiveness.

Participants expressed mixed feelings about introducing the HSCP intervention in the ED within a randomised controlled trial, appreciating the positive impact of the rigorous method on the profile of the intervention, but at the same time acknowledging some frustration among patient and staff due to the randomisation and trial procedures. At a contextual level, it was felt that the success of the implementation would not have been possible without the support and collaboration of invested and motivated stakeholders in the hospital and the broader healthcare setting, who facilitated the introduction of the HSCP team in the ED both at an operational and relational level.

### Results in the context of existing evidence

Although stakeholders' perspectives on HSCP services in the ED have been increasingly investigated [11–13, 17, 41], comparisons with previous literature are limited by the absence of a previous process evaluation of this model of care in the literature. Nonetheless, our findings on the implementation and mechanisms of impact of this intervention resonate with the available evidence in terms of positive views on HSCPs in the ED [17, 41].

With regards to the implementation process, the reported importance of having a pre-implementation and piloting period gathering inputs from multiple stakeholders, together with the perceived benefits of stakeholders' involvement and collaboration (Theme 4), support previous studies [24, 40] that have reported benefits of consultations and building support among key stakeholders before implementing a new model of ED care.

In line with existing evidence [11, 12, 40], this process evaluation supports the appropriateness of focusing allied health services for older adults in the ED on providing comprehensive assessment and intervention to support safe discharges. Our participants' concerns about the limitations of the HSCP team's operating hours and the need for extended services have been

expressed by both ED service users and providers in several studies [11, 12, 24, 41], and it was observed in a qualitative investigation that was conducted to inform the design of the present HSCP intervention [17]; however, the evidence on the impact of ED operating hours on patient outcomes is unclear [42, 43], and changing service hours requires a considerable revision of the scope of practice of both acute allied health and community services.

Our findings on the HSCP team's relational, professional, and interdisciplinary skills as key enablers of the successful implementation of the intervention, as well as a positive ED experiences for patients and staff, resonate with extensive evidence on the positive impact of teamwork, trustworthiness and proactivity within the ED on acceptability, buy-in and effectiveness of new models of care, particularly when changes in the culture of care are required [11–14, 41, 44, 45]. These findings point at the benefits of having a team of professionals with high motivation, a strong interdisciplinary skillset, and a patient-centred approach for the implementation of a collaborative intervention in a complex and dynamic environment such as the ED.

Despite these benefits, our study resonates with previous studies suggesting that interdisciplinary allied health interventions for older adults in the ED would be further optimised if providing care integration and community follow-ups together with a more advanced scope of practice for HSCPs and the support of geriatric specialist services [4, 46–49], although previous investigations have noted potential issues related to funding and sustainability [49]. More broadly, the logistic and operational issues associated with introducing a new model of care to a complex healthcare setting within randomised controlled trial have been acknowledged in previous research [50, 51], pointing at the need to consider more pragmatic methods [42].

## Strengths and limitations

To the best of our knowledge, this is the first structured process evaluation of a dedicated HSCP team providing early assessment and intervention to older adults in the ED. In this process evaluation, we adopted a standardised framework [23] and sought inputs from an interdisciplinary steering group of experts in allied health and emergency care to carry out a rigorous and comprehensive investigation of the multiple factors influencing the development and delivery of a new model of care in a complex healthcare environment. Using a mixed-methods approach enabled us to capture information both on the quality and quantity of the intervention; also, conducting both group and individual interviews helped our participants' recruitment as well as enriching our data collection.

The findings of the study should be considered in light of some limitations. Firstly, the process evaluation did not involve ED patients or carers, as their perceptions of the intervention have been investigated via a survey in the clinical effectiveness study [29] and it was felt that involving patients also in the process evaluation may cause an additional burden on potentially vulnerable individuals; nonetheless, we acknowledge that using accounts from staff members may have affected the accuracy of our findings on patients' experiences. Furthermore, exploring the views of community stakeholders (not involved in this study given the focus on the ED experience) could have enriched our understanding on the contribution of the intervention to patients' safe discharges. Our considerations on the implementation fidelity of the intervention could not be quantified due to the lack of an appropriate tool for this kind of intervention. Also, the generalisability of our findings may be limited given the nature of the intervention and the setting; contextual influences within and outside the ED may impact the implementation of this type of intervention elsewhere. Nonetheless, we found that the characteristics of intervention, as well as the mediators of impact, were in line with existing practice and evidence.

### Implications for practice

The insights gathered in this study suggest that a HSCP team intervention aimed at older adults in the ED is a viable strategy to improve the experiences of patients and staff members, provided adequate time for preparation and the recruitment of skilled and motivated professionals who are capable of building positive relationships and foster an interdisciplinary and patient-centred ethos in their practice. Furthermore, the active involvement of relevant stakeholders in the pre-implementation phase is essential to facilitate acceptance and the intervention operations. Our findings support the added value of introducing an interdisciplinary team of HSCPs over individual professionals for enhancing the of older adults with complex needs, because promoting collaborative decision-making and integrated care. Although funding and financial factors were not the focus of this study, understanding the economic costs and benefits associated with an interdisciplinary team-based model of care for this population is key to guide future implementation. A cost effectiveness evaluation is currently in preparation which will provide an account of this.

## Conclusions

Process evaluations serve to shed light on the pragmatics of implementing complex health service interventions [27]. This mixed-methods process evaluation described the multiple stages needed, and the stakeholder collaboration required, to develop and deliver an interdisciplinary HSCP intervention for older adults in the ED. We found that the positive relational and organisational skills of the HSCP team, together with their interdisciplinary ethos, were key determinants of the positive implementation of the intervention and its impact on patient and staff experiences, whereas aspects related to the trial format of the intervention caused operational constraints due to the pragmatics of the ED.

## Supporting information

**S1 Table. STARI checklist.** Standard reporting checklist.
(DOCX)

**S1 File. Implementation notes.** Detailed narrative analysis of implementation process and delivery.
(DOCX)

**S2 File. Theme quotes.** List of quotes from each theme.
(DOCX)

## Author Contributions

**Conceptualization:** Marica Cassarino, Úna Cronin, Katie Robinson, Rosie Quinn, Fiona Boland, Marie E. Ward, Rosa McNamara, Margaret O'Connor, Gerard McCarthy, Damien Ryan, Rose Galvin.

**Data curation:** Marica Cassarino.

**Formal analysis:** Marica Cassarino, Katie Robinson, Rose Galvin.

**Funding acquisition:** Katie Robinson, Rosie Quinn, Fiona Boland, Marie E. Ward, Rosa McNamara, Gerard McCarthy, Damien Ryan, Rose Galvin.

**Investigation:** Marica Cassarino, Rosie Quinn, Fiona Boland, Marie E. Ward, Rosa McNamara, Margaret O'Connor, Gerard McCarthy, Damien Ryan, Rose Galvin.

**Methodology:** Marica Cassarino, Úna Cronin, Katie Robinson, Margaret O'Connor, Rose Galvin.

**Project administration:** Úna Cronin, Damien Ryan.

**Resources:** Rose Galvin.

**Supervision:** Rose Galvin.

**Writing – original draft:** Marica Cassarino, Rose Galvin.

**Writing – review & editing:** Marica Cassarino, Úna Cronin, Katie Robinson, Rosie Quinn, Fiona Boland, Marie E. Ward, Rosa McNamara, Margaret O'Connor, Gerard McCarthy, Damien Ryan, Rose Galvin.

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
