## [Decision Letter · Decision Letter 0]

8 Mar 2022

PONE-D-21-31608Development and Delivery of an Allied Health Team Intervention for Older Adults in the Emergency Department: A Process EvaluationPLOS ONE

Dear Dr. Cassarino,

Thank you for submitting your manuscript to PLOS ONE. After careful consideration, we feel that it has merit but does not fully meet PLOS ONE’s publication criteria as it currently stands. Therefore, we invite you to submit a revised version of the manuscript that addresses the points raised during the review process.

We look forward to receiving your revised manuscript.

Kind regards,

Jason Scott

Academic Editor

PLOS ONE

Journal Requirements:

Reviewers' comments:

Reviewer's Responses to Questions

**Comments to the Author**

1. Is the manuscript technically sound, and do the data support the conclusions?

Reviewer #1: Yes

Reviewer #2: Yes

2. Has the statistical analysis been performed appropriately and rigorously? 

Reviewer #1: N/A

Reviewer #2: N/A

3. Have the authors made all data underlying the findings in their manuscript fully available?

Reviewer #1: No

Reviewer #2: Yes

4. Is the manuscript presented in an intelligible fashion and written in standard English?

Reviewer #1: Yes

Reviewer #2: Yes

5. Review Comments to the Author

Reviewer #1: Overall:

I really enjoyed reading this manuscript that focused on a process evaluation of a HSCP intervention

in an ED department. Overall, the paper is off good quality and will be interesting for the audience.

There are some instances where some more information would be beneficial which I have

highlighted in the comments. Thank you for submitting the manuscript.

Abstract:

Pg 3 – “This is the first process evaluation of the development and delivery of a HSCP team

intervention for older adults in the ED of a large Irish hospital.” – Could this be amended to form a

clear aim?

Pg 3 ln 45 – “employed activity and recruitment logs,” – of what specifically?

Pg 3 ln 45/46- “interviews and focus groups with 26 participants (HSCP team members, ED doctors

and nurses, hospital staff)” – can you differentiate here how many interviews and how many focus

groups were conducted?

Introduction:

Pg 4 ln 76/77 – “patients with complex needs” – could you provide some examples?

Pg 6 – Thank you for describing the three main objectives, could you provide an overall aim of the

evaluation please?

Materials and methods:

Design: Thank you for clearly documenting that the evaluation has been reported in line with the

Standards for Reporting Implementation Studies (STARI) statement.

Context:

Pg 7 – Could you provide further rationale here for why the target population for the intervention

were “frail older adults aged ≥65 years”. You briefly mention this on pg 12 under the theme

‘Intervention delivery: fidelity, dose, and reach’, however this would be beneficial for the reader to

know.

Participants:

Pg 7 – Could you indicate which participants took part in the 1:1 interview and who took part in the

focus groups? For instance, did all the ED nurses and ED doctors take part in a focus group?

Pg 7 - I also don’t think the specific role of the participants are relevant and you could just leave as

“three HSCPs, 10 ED doctors, 10 ED nursing staff members, three hospital staff members”.

Pg 8 ln 153-154 – You have already started that participants were required to sign a consent form in

line 151, please remove one of the instances.

Data collection and analysis:

Pg 9 – Given in the participants that you indicate that there are six 1:1 interview, can you indicate

how many focus groups were conducted please and how many people were involved in each?

Pg 9 ln 163-166 – Were these definitions that the research team developed or amended from

elsewhere – please state.

Pg 9 ln 171 – Can you provide rationale as to why focus groups and interviews were used? Instead of

just one or the other?

Pg 9 ln 173 – “quantitative and descriptive data on implementation...” – quantitative and descriptive

data on what specifically?

Pg 9 ln 173 – "Qualitative data was integrated with quantitative" - Can you provide further explanation how you achieved this and the methods you used?

Pg 9 ln 176 – need a space after the protocol reference

Pg 10 ln 182-183 – “Data saturation was agreed by the two researchers (MC and RG) (34).” – Sorry if

I have misunderstood this, does this them mean you were aiming to recruit more than 26 people in

the first instance, but then stopped due to data saturation? If this is the case then can you please

provide this detail.

Results:

Pg 11 ln 210 – “supports needed for the successful implementation” – does not make sense

Pg 13 ln 249 – “a lot of eligible patients were GP referrals” – can you be more specific please?

Pg 13 ln 254 – “discharge the patient safely:” – full stop?

Pg 13 ln 255 – “(65+)” - change to ≥65 years

Pg 14 Table 3. Intervention delivery – One section has a full stop and the other do not. Can you

amend to be consistent please. Also, the size font in ‘Reach’ seems to be larger than the other

sections.

Pg 15 ln 275 & 277 – Swap the “;” and “.” for point 3 and 4

Themes – Can you provide a table of quotes for the four themes? This will highlight the range of

perspectives and quotes from the participants. Then signpost the reader to the quotes for the table

of quotes.

Themes – Given the number of participants who had the same job, when providing quotes can you

indicate whether they are the same or different participants? Possibly put (ED Doctor 1), (ED Doctor

2) etc. This will be useful for the reader to show the perspectives are from a range of participants.

Pg 16 ln 306 – “(HSCP team member)” – Brackets in italics

Discussion:

Overall I think the discussion is well written.

Pg 23 ln 464 – “wouldn’t” – Change to “would not”

Pg 23 ln 469-472 – I feel that the first paragraph is a strength, as opposed contextualising the results

in line with the literature. I would recommend moving this to the later section and starting the

section from “Out findings on the…”

Reviewer #2: Overall comments

This is a generally well written article that provides a process evaluation related to the introduction of an interdisciplinary team of Health and Social Care Professionals (HSCPs) to an ED in Ireland. While it is not a novel concept to include HSCPs in ED making the model of care work efficiently and effectively is often elusive and more research needs to be undertaken in this space. Process evaluation alongside an RCT is excellent research practice in this space.

Abstract

Clear and complete

Introduction

This section provides a clear background and rationale to the study.

Methods

The method section is generally clear and contains all the relevant information, albeit through reference to two previously published articles. It is disappointing that this process evaluation did not involve patients and/or their carers as most work in this area tries to also present a consumer perspective. However, I understand how difficult this can be with this very vulnerable population.

One aspect I would question, given that much of the data related to interviews/focus groups and was therefore qualitative rather than quantitative data, is the use of the heading “Measures” (line 155). The content of this section includes information about the data collected and how the type of data related to the study objectives. Most data are not measuring something but rather describing and exploring concepts and experiences. I would suggest a change to the heading and table heading.

Under the heading context information is provided about the ED but the context of this implementation study is also about the RCT and the intervention being tested. I suggest providing some information here about the intervention (even though it is detailed in other articles it would help the reader to have some basic information here). Later in the article it says the intervention targeted “frail” (line 227) older adults – there is no indication about how this was assessed prior to participants’ inclusion in the trial. The Trial report states, “Prospective patients who had a Manchester Triage System score (MTS) of 3 to 5 (respectively, urgent, standard, or nonurgent) and who were categorised at triage with limb problems, falls, unwell adult, back pain, urinary problems, or ear/facial problems were invited to take part.” This does not include a frailty assessment. The Rockwood Clinical Frailty Scale was completed after consent was obtained. If a frailty assessment was completed before recruitment as part of a screening process I suggest including this information in this article. This information is important as it allows others to assess the transferability of these findings.

One minor point – “data” is the plural form of the noun “datum” and should be linked with the plural version of the verb e.g. “data were…”. Please change throughout.

Results

Again, with qualitative studies a more usual heading is “Findings” as this report is not the result of some experiment.

Minor point – sentence in lines 243-245 may need revision the English expression is poor.

I am concerned that this process evaluation contradicts the trial report in relation to patient consent for the RCT. In this article it is reported that due to the research nurse’s holiday the intervention HSCPs consented trial participants. As this was not reported in the trial results article, and because this could significantly affect how free potential participants might have felt about declining an invitation to participate, I believe this needs further clarification. Specifically, the authors need to indicate that this variation to the protocol was reported to and approved by the ethics committee that gave trial approval.

Discussion

The comment in the section on Limitations “Lastly, while every effort was made 530 to ensure rigour, potential respondent and researcher's biases associated with qualitative data 531 collection cannot be entirely excluded.” Does not demonstrate a good understanding of the issues of rigour in qualitative studies. I suggest either including a much more comprehensive statement about how this research demonstrated rigour – or because you don’t have the space – deleting this sentence.

I understand that this intervention was trialled in an ED funded by public funds, however, in my experience a major aspect of the success of such interventions is whether recurrent funding can be accessed once the trial is completed. I am surprised there was no discussion of the issues associated with funding. I note that an economic evaluation was not included in the main trial. I believe a comment about this, at least, in the Discussion or as a recommendation for further study might be useful.

The difficulty with getting such programs funded is that if these complex, vulnerable patients spend less time in ED (which is a great outcome for the individuals) the ED beds rarely stay empty. As a consequence, the cost of providing ED services does not decrease. Rather, issues such as overcrowding, ambulance ramping etc may reduce but rarely cost. Arguing this is an important part of securing long term senior management support and on-going funding. Again, a comment in the discussion might be helpful here.

6. PLOS authors have the option to publish the peer review history of their article (what does this mean?). If published, this will include your full peer review and any attached files.

Reviewer #1: No

Reviewer #2: **Yes: **Marianne Wallis

---

## [Author Response · Author response to Decision Letter 0]

31 Mar 2022

Editorial Office

E1. Please ensure that your manuscript meets PLOS ONE's style requirements, including those for file naming. The PLOS ONE style templates can be found at 

Response – We have revised the format of the manuscript based on the guidance provided. We have edited the file names for the supporting information. If any further formatting changes are required, please do not hesitate to point them out to us. 

E2. In your Data Availability statement, you have not specified where the minimal data set underlying the results described in your manuscript can be found. PLOS defines a study's minimal data set as the underlying data used to reach the conclusions drawn in the manuscript and any additional data required to replicate the reported study findings in their entirety. All PLOS journals require that the minimal data set be made fully available. For more information about our data policy, please see http://journals.plos.org/plosone/s/data-availability. Upon re-submitting your revised manuscript, please upload your study’s minimal underlying data set as either Supporting Information files or to a stable, public repository and include the relevant URLs, DOIs, or accession numbers within your revised cover letter. For a list of acceptable repositories, please see http://journals.plos.org/plosone/s/data-availability#loc-recommended-repositories. Any potentially identifying patient information must be fully anonymized. Important: If there are ethical or legal restrictions to sharing your data publicly, please explain these restrictions in detail. Please see our guidelines for more information on what we consider unacceptable restrictions to publicly sharing data: http://journals.plos.org/plosone/s/data-availability#loc-unacceptable-data-access-restrictions. Note that it is not acceptable for the authors to be the sole named individuals responsible for ensuring data access. We will update your Data Availability statement to reflect the information you provide in your cover letter.

A. Please name the institution who has imposed these restrictions upon your data (e.g., a Research Ethics Committee or Institutional Review Board, etc.). Was it the Health Service Executive (HSE) Mid-Western Regional Hospital Research Ethics Committee?

B. Please note that it is not acceptable for authors of the study to be either the sole listed point of contact for data or the ultimate provider of data that is made available upon request. At this time, please provide direct, non-author contact information (preferably email) for the body imposing the restrictions upon the data or to a person who may grant the data after permission is given, who is not the corresponding author.

Response – Thank you for this comment. Our study data falls within the category of Human research participant data with restrictions applying. As requested, we have updated the Data Availability Statement to a) clarify that the Health Service Executive (HSE) Mid-Western Regional Hospital Research Ethics Committee has imposed these restrictions on the sharing of data pertaining to this study; b) Data are available from the Health Service Executive (HSE) Mid-Western Regional Hospital Research Ethics Committee (ULHGResearchEthicsandClinicalTrials@hse.ie) for researchers who meet the criteria for access to confidential data.

In line with the restrictions imposed by the Health Service Executive (HSE) Mid-Western Regional Hospital Research Ethics Committee, which approved the ethics of this study, data collected in this study cannot be shared publicly because participants can be identifiable given their role in the setting of the study, and because the participants did not give consent to have their responses made publicly available. Data are available from the Health Service Executive (HSE) Mid-Western Regional Hospital Research Ethics Committee (ULHGResearchEthicsandClinicalTrials@hse.ie) for researchers who meet the criteria for access to confidential data. 

E3. Please review your reference list to ensure that it is complete and correct. If you have cited papers that have been retracted, please include the rationale for doing so in the manuscript text, or remove these references and replace them with relevant current references. Any changes to the reference list should be mentioned in the rebuttal letter that accompanies your revised manuscript. If you need to cite a retracted article, indicate the article’s retracted status in the References list and also include a citation and full reference for the retraction notice.

Response – We have reviewed the reference list and updated any information that appeared to be incorrect. We confirm that none of the included references has been retracted. One reference (39) was edited to include the new url where it can be accessed. 

Reviewer 1

R1C1: Overall: I really enjoyed reading this manuscript that focused on a process evaluation of a HSCP intervention in an ED department. Overall, the paper is off good quality and will be interesting for the audience. There are some instances where some more information would be beneficial which I have highlighted in the comments. Thank you for submitting the manuscript.

Response – Thank you for your positive and constructive feedback. We hope that our revisions have improved the quality of the manuscript. Please note that page and line numbers refer to the marked-up copy of the revised manuscript. 

R1C2: Abstract Pg 3 – “This is the first process evaluation of the development and delivery of a HSCP team intervention for older adults in the ED of a large Irish hospital.” – Could this be amended to form a clear aim?

Response – We have amended this to read as follows (p.3, l.43-45): “The study aimed to evaluate the development and delivery of a HSCP team intervention for older adults in the ED of a large Irish teaching hospital”

R1C3 : Abstract Pg 3 ln 45 – “employed activity and recruitment logs,” – of what specifically?

Response – We have specified that we analysed “the HSCP team’s activity notes and participant recruitment logs” (p.3 l.49-50)

R1C4: Abstract Pg 3 ln 45/46- “interviews and focus groups with 26 participants (HSCP team members, ED doctors and nurses, hospital staff)” – can you differentiate here how many interviews and how many focus groups were conducted?

Response – We have clarified at p.3 that we carried out “six interviews and four focus groups with 26 participants” (l.50-51)

-

R1C5: Intro Pg 4 ln 76/77 – “patients with complex needs” – could you provide some examples?

Response – We have added (p.4 l.82) “with complex health and social care needs including older adults and those with multimorbidity”. 

R1C6: Intro Pg 6 – Thank you for describing the three main objectives, could you provide an overall aim of the evaluation please?

Response – We have clarified (p.6 l.113-114) that “The overall aim of this process evaluation is to provide a comprehensive account of the development and delivery of this ED-based HSCP intervention, and it has three main objectives”

R1C7: Design: Thank you for clearly documenting that the evaluation has been reported in line with the Standards for Reporting Implementation Studies (STARI) statement.

Response – Thank you for your positive feedback

R1C8: Context: Pg 7 – Could you provide further rationale here for why the target population for the intervention were “frail older adults aged ≥65 years”. You briefly mention this on pg 12 under the theme ‘Intervention delivery: fidelity, dose, and reach’, however this would be beneficial for the reader to know.

Response – We have clarified now at p.7 l.142-147 that “Older adults represent approximately 30% of presentations in this setting (32) and are more likely to present with a complexity of functional, medical and psychosocial issues. This cohort were identified as a target group who were most likely to benefit the most from this model of care in our extensive stakeholder engagement meetings prior to undertaking the RCT [17].”

R1C9: Participants Pg 7 – Could you indicate which participants took part in the 1:1 interview and who took part in the focus groups? For instance, did all the ED nurses and ED doctors take part in a focus group?

Response – We have now specified at p.8 l.165-167 that “The four focus groups comprised the HSCP team, eight ED doctors, and nine nurses. The research nurse involved in the trial, two ED doctors, and the three hospital staff members completed 1:1 interviews”.

R1C10: Participants Pg 7 - I also don’t think the specific role of the participants are relevant and you could just leave as “three HSCPs, 10 ED doctors, 10 ED nursing staff members, three hospital staff members”.

Response – We thank the Reviewer for this point. We have simplified the description of ED doctors and nurses but kept the description of hospital staff members to highlight their role. (see p.8)

R1C11: Participants Pg 8 ln 153-154 – You have already started that participants were required to sign a consent form in line 151, please remove one of the instances.

Response – Thank you for noting this error. The second instance of this sentence has been removed (p.9 l.173-174). 

R1C12: Data collection and analysis: Pg 9 – Given in the participants that you indicate that there are six 1:1 interview, can you indicate how many focus groups were conducted please and how many people were involved in each?

Response – We have clarified this in the participants section, see p.8 l.153 and l.165-167

R1C13: Pg 9 ln 163-166 – Were these definitions that the research team developed or amended from elsewhere – please state.

Response – We have clarified at p.9 l.182-183 that the “definitions were developed and agreed by the research team”

R1C14: Pg 9 ln 171 – Can you provide rationale as to why focus groups and interviews were used? Instead of just one or the other?

Response – We have clarified in the Data collection and analysis section (p.10 l.194-196) that “Using both focus groups and interviews enabled better accommodation of participants’ needs with regards to their availability and their preference for sharing their experience with others or individually”

R1C15: Pg 9 ln 173 – “quantitative and descriptive data on implementation...” – quantitative and descriptive data on what specifically?

Response – We have now amended the sentence to the following (p.10 l.196-199): “Qualitative data regarding the development and delivery of the intervention were integrated with quantitative data and notes from the recruitment and activity logs to develop a description of fidelity, dose, and reach”

R1C16: Pg 9 ln 173 – "Qualitative data was integrated with quantitative" - Can you provide further explanation how you achieved this and the methods you used?

Response – We have amended this to the following: “Qualitative data regarding the development and delivery of the intervention were integrated with quantitative data and notes from the recruitment and activity logs to develop a description of fidelity, dose, and reach” (p.10 l.196-199). This is evident, for instance, in the Intervention fidelity section, which was developed taking into account both the team’s notes and recruitment logs, as well as their qualitative inputs during the focus group. 

R1C17: Pg 9 ln 176 – need a space after the protocol reference

Response – Thank you. This has been edited (p.10 l.200)

R1C18: Pg 10 ln 182-183 – “Data saturation was agreed by the two researchers (MC and RG) (34).” – Sorry if I have misunderstood this, does this them mean you were aiming to recruit more than 26 people in the first instance, but then stopped due to data saturation? If this is the case then can you please provide this detail.

Response – Thank you for allowing us to clarify this point. Yes, we adopted data saturation, meaning that we would proceed with recruitment until the researchers involved in the analyses felt that responses converged around the same categories. In doing so, we were guided by Saunders et al’s classification of saturation. This was possible because the responses were analysed iteratively. We have now specified this at p.11 l.208 and we have clarified that “In line with Saunders et al (34), data saturation was agreed by the two researchers (MC and RG) during data collection when it was observed that new data repeated what had been already expressed in previous responses” (p.11, l209-211)

R1C19: results Pg 11 ln 210 – “supports needed for the successful implementation” – does not make sense

Response – We have amended this paragraph to increase readability. The sentence noted by the reviewer has been edited to the following (p.12 l.236-238): “to determine the structures and the supports needed for a successful implementation”

R1C20: Pg 13 ln 249 – “a lot of eligible patients were GP referrals” – can you be more specific please?

Response – Thank you for this comment. We appreciate the vagueness of this statement and acknowledge that this was reported anecdotally by the HSCP team. A quantitative analysis of this aspect would be beyond the scope of the trial data, thus, we have amended this sentence to reflect that this is an anecdotal account (p.14, l.277-278): “anecdotally the HSCP team members reported that many eligible patients were GP referrals presenting in the afternoon”. 

R1C21: Pg 13 ln 254 – “discharge the patient safely:” – full stop?

Response – Thank you. This has been fixed (p.14 l.282)

R1C22: Pg 13 ln 255 – “(65+)” - change to ≥65 years

Response – This has been amended as suggested (p.14 l.283). 

R1C23: Pg 14 Table 3. Intervention delivery – One section has a full stop and the other do not. Can you amend to be consistent please. Also, the size font in ‘Reach’ seems to be larger than the other sections.

Response – The full stops have been removed and the font size has been fixed (Table 3, p.15). 

R1C24: Pg 15 ln 275 & 277 – Swap the “;” and “.” for point 3 and 4

Response – This has been fixed (p.15 l.304 and l.306)

R1C25: Themes – Can you provide a table of quotes for the four themes? This will highlight the range of perspectives and quotes from the participants. Then signpost the reader to the quotes for the table of quotes.

Response – We thank the Reviewer for this suggestion. While we agree that having an overview of the quotes may be useful, we feel that having a table may cause either repetition with the quotes already presented in the main body, or impair the narrative developed across the four sections related to the themes. To this end, we have gathered the quotes in a new S2 File and refer the reader to the supporting documentation on p.16 l.307. 

R1C26: Themes – Given the number of participants who had the same job, when providing quotes can you indicate whether they are the same or different participants? Possibly put (ED Doctor 1), (ED Doctor 2) etc. This will be useful for the reader to show the perspectives are from a range of participants.

Response – Participants have been numbered across the Results section as suggested. 

R1C27: Pg 16 ln 306 – “(HSCP team member)” – Brackets in italics

Response – This has been fixed (p.17 l.335)

R1C28: Discussion - Overall I think the discussion is well written. Pg 23 ln 464 – “wouldn’t” – Change to “would not”

Response – This has been fixed (p.24 l.494). Thank you for your positive feedback on the Discussion section. 

R1C29: Pg 23 ln 469-472 – I feel that the first paragraph is a strength, as opposed contextualising the results in line with the literature. I would recommend moving this to the later section and starting the section from “Out findings on the…”

Response – We thank the Reviewer for this point. We have moved part of the sentence to the strengths section, p.26 l.540-541. We have rephrased the suggested sentence to highlight that the absence of a previous process evaluation for this model of care limits our ability to make comprehensive comparisons (p.24 l.499-503): “Although stakeholders’ perspectives on HSCP services in the ED have been increasingly investigated (11–13,17,41), comparisons with previous literature are limited by the absence of a previous process evaluation of this model of care in the literature”

Reviewer 2

R2C1: This is a generally well written article that provides a process evaluation related to the introduction of an interdisciplinary team of Health and Social Care Professionals (HSCPs) to an ED in Ireland. While it is not a novel concept to include HSCPs in ED making the model of care work efficiently and effectively is often elusive and more research needs to be undertaken in this space. Process evaluation alongside an RCT is excellent research practice in this space.

Response – We thank the Reviewer for the positive feedback and hope that the revisions outlined below have enhanced the clarity and quality of the paper. Please note that page and line numbers refer to the marked-up copy of the revised manuscript. 

R2C2: Abstract - Clear and complete

Response – Thank you for the positive comment. 

R2C3: Introduction - This section provides a clear background and rationale to the study.

Response – Thank you for the positive comment. 

R2C4 - The method section is generally clear and contains all the relevant information, albeit through reference to two previously published articles. It is disappointing that this process evaluation did not involve patients and/or their carers as most work in this area tries to also present a consumer perspective. However, I understand how difficult this can be with this very vulnerable population.

Response – We thank the Reviewer for the comments, and we agree that involving patients and carers would have certainly enhanced the richness of our data. We note this in our limitations section, p.26 l.549-557. We have made some revisions to the methods section (see context, participants, and data collection) that have hopefully increased clarity. 

R2C5 - One aspect I would question, given that much of the data related to interviews/focus groups and was therefore qualitative rather than quantitative data, is the use of the heading “Measures” (line 155). The content of this section includes information about the data collected and how the type of data related to the study objectives. Most data are not measuring something but rather describing and exploring concepts and experiences. I would suggest a change to the heading and table heading.

Response – We thank the Reviewer for this point and agree that measures may not be fully appropriate to this section and indeed this study. We have edited the heading “Measures” to “Explored domains” and changed the word “measure” for “domain” where appropriate in the section and table 1 (see p.9 l.175 to Table 1). 

R2C6: Under the heading context information is provided about the ED but the context of this implementation study is also about the RCT and the intervention being tested. I suggest providing some information here about the intervention (even though it is detailed in other articles it would help the reader to have some basic information here). Later in the article it says the intervention targeted “frail” (line 227) older adults – there is no indication about how this was assessed prior to participants’ inclusion in the trial. The Trial report states, “Prospective patients who had a Manchester Triage System score (MTS) of 3 to 5 (respectively, urgent, standard, or nonurgent) and who were categorised at triage with limb problems, falls, unwell adult, back pain, urinary problems, or ear/facial problems were invited to take part.” This does not include a frailty assessment. The Rockwood Clinical Frailty Scale was completed after consent was obtained. If a frailty assessment was completed before recruitment as part of a screening process I suggest including this information in this article. This information is important as it allows others to assess the transferability of these findings.

Response – Thank you for this suggestion. We have now included a brief description of the intervention as follows (p.7, l.147-151): “The intervention consisted of a dedicated team of HSCPs providing early assessment and intervention to older people aged ≥65 years presenting to the ED with complex care needs. The impact of the intervention when compared to usual care was tested on our two primary outcomes of ED length of stay and hospital admission rates. Secondary outcomes focused on a range of other patient, process, and clinical outcomes.“

We agree with the Reviewer that the term “frail” is not appropriate in this context because not based on a formal assessment. Thus, we have removed this term where appropriate (p.13 l.254; p.14 L.284)

R2C7: One minor point – “data” is the plural form of the noun “datum” and should be linked with the plural version of the verb e.g. “data were…”. Please change throughout.

Response – We have fixed this at p.10 l.197, p.11 l.208

R2C8 - Results Again, with qualitative studies a more usual heading is “Findings” as this report is not the result of some experiment.

Response – We have amended the heading as suggested (p.11 l.222)

R2C9 Results Minor point – sentence in lines 243-245 may need revision the English expression is poor.

Response – We have amended the sentence to read as follows (p.14 l.269-273): “These adaptations helped the team to share with the medical team a more comprehensive analysis and clearer recommendations for the patients, and thus enabled more timely care decisions.”

R2C10 - I am concerned that this process evaluation contradicts the trial report in relation to patient consent for the RCT. In this article it is reported that due to the research nurse’s holiday the intervention HSCPs consented trial participants. As this was not reported in the trial results article, and because this could significantly affect how free potential participants might have felt about declining an invitation to participate, I believe this needs further clarification. Specifically, the authors need to indicate that this variation to the protocol was reported to and approved by the ethics committee that gave trial approval.

Response – Thank you for this point. We confirm that this was an amendment to the initial protocol that received ethical approval in November 2018. We have clarified this in the main body, p.22 l.441-442: “These adaptations received full ethical approval before being implemented”

R2C11 - Discussion The comment in the section on Limitations “Lastly, while every effort was made 530 to ensure rigour, potential respondent and researcher's biases associated with qualitative data 531 collection cannot be entirely excluded.” Does not demonstrate a good understanding of the issues of rigour in qualitative studies. I suggest either including a much more comprehensive statement about how this research demonstrated rigour – or because you don’t have the space – deleting this sentence.

Response – This sentence has been deleted (p.27 l.563-565). 

R2C12 - I understand that this intervention was trialled in an ED funded by public funds, however, in my experience a major aspect of the success of such interventions is whether recurrent funding can be accessed once the trial is completed. I am surprised there was no discussion of the issues associated with funding. I note that an economic evaluation was not included in the main trial. I believe a comment about this, at least, in the Discussion or as a recommendation for further study might be useful.

Response – Thank you for this point. We have now clarified in the Introduction (p.6 l.111-112) and in the Discussion that a cost effectiveness evaluation is currently in preparation as a separate publication and that that will focus on the economic dimension of the intervention (see p.27 l.576-579): “Although funding and financial factors were not the focus of this study, understanding the economic costs and benefits associated with an interdisciplinary team-based model of care for this population is key to guide future implementation. A cost effectiveness evaluation is currently in preparation which will provide an account of this.”

R2C13 - The difficulty with getting such programs funded is that if these complex, vulnerable patients spend less time in ED (which is a great outcome for the individuals) the ED beds rarely stay empty. As a consequence, the cost of providing ED services does not decrease. Rather, issues such as overcrowding, ambulance ramping etc may reduce but rarely cost. Arguing this is an important part of securing long term senior management support and on-going funding. Again, a comment in the discussion might be helpful here.

Response – We agree with the Reviewer and have now added a note on this at p.25 l.533-535 where we discuss previous evidence on care coordination, which point at potential issues with sustainability and funding: 

“[4,47–50], although previous investigations have noted potential issues related to funding and sustainability [50]”

As we note in the new paragraph added to the implications for practice (see response to comment R2C12), the cost evaluation will provide further information on the costs and benefits associated with this type of intervention.

---

## [Decision Letter · Decision Letter 1]

16 May 2022

Development and Delivery of an Allied Health Team Intervention for Older Adults in the Emergency Department: A Process Evaluation

PONE-D-21-31608R1

Dear Dr. Cassarino,

We’re pleased to inform you that your manuscript has been judged scientifically suitable for publication and will be formally accepted for publication once it meets all outstanding technical requirements.

Kind regards,

Jason Scott

Academic Editor

PLOS ONE

Additional Editor Comments (optional):

Reviewers' comments:

Reviewer's Responses to Questions

**Comments to the Author**

1. If the authors have adequately addressed your comments raised in a previous round of review and you feel that this manuscript is now acceptable for publication, you may indicate that here to bypass the “Comments to the Author” section, enter your conflict of interest statement in the “Confidential to Editor” section, and submit your "Accept" recommendation.

Reviewer #1: All comments have been addressed

Reviewer #2: (No Response)

2. Is the manuscript technically sound, and do the data support the conclusions?

Reviewer #1: Yes

Reviewer #2: (No Response)

3. Has the statistical analysis been performed appropriately and rigorously? 

Reviewer #1: N/A

Reviewer #2: (No Response)

4. Have the authors made all data underlying the findings in their manuscript fully available?

Reviewer #1: (No Response)

Reviewer #2: (No Response)

5. Is the manuscript presented in an intelligible fashion and written in standard English?

Reviewer #1: Yes

Reviewer #2: (No Response)

6. Review Comments to the Author

Reviewer #1: I want to congratulate the authors in making the amendments and resubmitting the manuscript. The authors clearly and articulately responded to my comments and suggestions, as well as signposted me to their amendment. This made reviewing the amendments much easy, so thank you. Overall, the additions to the paper from both reviewers have made great additions to the paper.

Reviewer #2: (No Response)

7. PLOS authors have the option to publish the peer review history of their article (what does this mean?). If published, this will include your full peer review and any attached files.

Reviewer #1: **Yes: **Dr Kate Byrnes

Reviewer #2: **Yes: **Marianne Wallis

---

## [Editor Report · Acceptance letter]

18 May 2022

PONE-D-21-31608R1 

Development and Delivery of an Allied Health Team Intervention for Older Adults in the Emergency Department: A Process Evaluation 

Dear Dr. Cassarino:

I'm pleased to inform you that your manuscript has been deemed suitable for publication in PLOS ONE. Congratulations! Your manuscript is now with our production department. 

Kind regards, 

on behalf of

Dr. Jason Scott 

Academic Editor

PLOS ONE